# Food Insecurity and Major Diet-Related Morbidities in Migrating Children: A Systematic Review

**DOI:** 10.3390/nu12020379

**Published:** 2020-01-31

**Authors:** Arianna Dondi, Valentina Piccinno, Francesca Morigi, Sugitha Sureshkumar, Davide Gori, Marcello Lanari

**Affiliations:** 1Pediatric Emergency Unit, Department of Medical and Surgical Sciences (DIMEC), S.Orsola Hospital, University of Bologna, 40138 Bologna, Italy; francesca.morigi@gmail.com (F.M.); marcello.lanari@unibo.it (M.L.); 2Pediatric and Neonatology Unit, Imola Hospital, 40026 Imola (Bologna), Italy; valentinapiccinno@gmail.com; 3Institute of Global Health, University of Geneva, 1205 Geneva, Switzerland; sugi84@hotmail.com; 4Department of Biomedical and Neuromotor Sciences, University of Bologna, 40100 Bologna, Italy; davide.gori4@unibo.it

**Keywords:** migration, children, vulnerable groups, food insecurity, obesity, stunting, vitamin D, iron, early childhood caries, diet

## Abstract

Children of migrant families are known to be at a higher risk of diet-related morbidities due to complex variables including food insecurity, cultural and religious beliefs, and sociodemographic factors like ethnicity, socioeconomic status, and education. Several studies have assessed the presence of specific diseases related to dietary issues in migrant children. This systematic review aims to highlight the existing body of work on nutritional deficiencies in the specific vulnerable pediatric population of immigrants. Refugees were intentionally excluded because of fundamental differences between the two groups including the reasons for migration and health status at the time of arrival. A total of 29 papers were included and assessed for quality. Most of them described a strong correlation between obesity and migration. A high prevalence of stunting, early childhood caries, iron and vitamin D deficiency was also reported, but the studies were few and heterogeneous. Food insecurity and acculturation were found important social factors (nevertheless with inconclusive results) influencing dietary habits and contributing to the development of morbidities such as obesity and other metabolic disorders, which can cause progressive unsustainability of health systems. Public health screening for diet-related diseases in migrant children may be implemented. Educational programs to improve children’s diet and promote healthy-living behaviors as a form of socioeconomic investment for the health of the new generations may also be considered.

## 1. Introduction

During the course of history migratory patterns have been defined by demands for work, education, escape from armed conflict and poverty, or climate change [1,2]. In the recent past, high-income countries (HIC) have consistently received international migrants with steady increases over time. The estimated number of migrants to Europe and Northern America exceeded the number of emigrants by 25.9 million in the decade 2010–2020 [1,2]. This paper does not make reference to the political aspects of migration surrounding voluntary and involuntary migration (the forced displacement precluding asylum seekers and refugees), and aims to objectively provide the status of nutrition amongst children in migrant populations globally.

Food insecurity (FI) is a common problem among migrant people and entails several issues such as cultural and religious beliefs, socio-demographic, economic, and environmental factors, education, and lifestyle changes [3,4]. It has been associated with a number of negative health and behavioral outcomes, mostly pertaining to children [5,6,7,8]. The most commonly referenced resource for the definition of Food Security is the US Department of Agriculture, that describes it as “access by all people at all times to enough food for an active, healthy life” [9]. Thus, the inverse proves true for FI.

Several studies point out that immigrant families, compared to non-immigrant ones, disproportionately experience FI, which is a concerning factor given the increasing size of this population [10,11]. Most of the studies about the relationship between FI and migration were conducted in the USA, but the growth rate of FI among children is also a well-known issue in European countries. Up to 18% of European households experience moderate-to-severe inabilities to access food and up to 20% report not enough money to buy food [12,13]. One of the causes that can be found in the loss of access to types of food that was previously familiar and known, with which they have a cultural and traditional connection. This factor, combined with a low socio-economic setting and a lack of knowledge about the nutritional values of food available in the host country, might contribute to the increase in FI in these populations. All these behaviors have been related to a number of negative health implications, particularly for children and adolescents that are the most vulnerable sub-population, with regards to the age-specific needs [4]. The lack of access to specific nutrients and to a varied and balanced diet, which are essential components for developing organisms, can lead to micronutrient deprivation and predispose to diseases such as anemia and rickets. Iron deficiency (ID) and low levels of vitamin D are common in migrant children in HIC [14,15]. Risk factors for vitamin D deficiency include being of female gender and having dark skin pigmentation. Inadequate levels of vitamin D prevent proper bone development, which can result in rickets in growing children and osteomalacia when growth has ended [15], while depletion of the iron stores can lead to anemia and, when it starts from early infancy and persists over time, even to neuro-cognitive delay [16].

Migration from low to high resource settings could also lead to FI without hunger, but with malnutrition. Indeed, economic and cultural factors could also lead to dietary simplifications with overconsumption of high energy, low-cost, nutrient-poor, sugary and fatty products and beverages (junk foods), which are directly linked to obesity, metabolic disorders, cardiovascular diseases [17], early childhood caries (ECC) [18,19]. The American Academy of Pediatric Dentistry defines ECC as “the presence of 1 or more decayed (non-cavitated or cavitated lesions), missing (due to caries), or filled tooth surfaces” in any primary tooth in a child 71 months of age or younger [20]. High simple-sugar diet, insufficient exposure to fluoride, poor oral hygiene and lack of financial resources to access dental care are implicated in the development of ECC, which eventually may lead to pain, abscesses, impaired speech, and eating problems [21].

This systematic review aims to highlight the existing body of work on nutritional deficiencies in the specific vulnerable pediatric population of immigrants with the exclusion of refugees. In fact, despite a frequent overlapping of the two terms, there are substantial differences between the two groups: immigrants normally choose to move from their original setting to a new one, whereas refugees are normally forced to move due to geopolitical reasons and are considered to require protection [22]; refugees commonly arrive in worse conditions and are at a higher risk of transitioning to poorer health [23,24,25]. For these reasons, refugees are seen here as a significantly different population and the study has been limited to immigrants. The research questions addressed include: (1) Is there documented evidence of a correlation between migration, FI, and morbidity amongst the pediatric population? (2) Are there any specified diseases with a clear association to dietary problems and food insecurity in migrant children?

## 2. Materials and Methods

This systematic literature review was performed in November 2019 by browsing into the following databases: Medline (Pubmed), Cochrane Library, Clinicaltrials.gov. Research was improved by searching in the most important websites of guidelines and clearinghouses and in the most important etextbooks sites.

The Preferred Reporting Items for Systematic Reviews and Meta-analyses guidelines (PRISMA) flow chart [26] was used to guide the transparent exclusion of published literature with defined reasons. The pertinent literature which was selected was hence tabulated with regards to country of origin, migration to, study design, number, age, and migrant characteristics, outcome disease, dietary issues, and main results. Further reporting surrounded bias, data reliability, and quality, to help the decision-making process during designing, implementing, and evaluating health intervention projects and policy.

The present work was registered to the international prospective register of systematic reviews, PROSPERO, with the following registration number: CRD42019145319.

Inclusion Criteria: Date: published between January 2005 and November 2019; Exposure: Migrant children and nutritional status; Population: migrants from low to high resource settings (different regions in the same country to be included); Language: English; Study Design: RCT, cohort studies, cross-sectional, retrospective; Outcomes: reported.

Exclusion Criteria: Exposure: not pertaining to migrant children and nutritional status; Population: refugees; Language: other than English; Study Design: Descriptive studies, reports, protocols; Outcomes: unreported.

Search String: ((migrant* OR transient* OR emigrant* OR immigra* OR refugee* OR migrant[MH]) AND (child OR children) AND (religion OR religio* OR Religious belie* OR religious eth* OR religious pract* OR cultural belie* OR socioeconomic factors OR education OR income OR feeding behaviour OR ethnicity) AND (food insecurity OR food security OR food policy OR food supply OR food insec* OR food sec* OR food poli* OR food suppl* OR processed food OR junk food OR western-style diet OR sugar sweetened beverages OR SSB OR soft drinks OR snack OR snacks OR fast food) AND (metabolic syndr* OR metabolic syndrome OR obes* OR obesity OR hypertension OR cardiovascular morbidity OR CV Morbidity OR malnutrition OR failure to thrive OR dental caries OR rickets OR iron deficiency OR iron defic* OR vitamin defic*)).

Quality Assessment: The Effective Public Health Practice Project (EPHPP) Quality assessment tool [27] for quantitative studies was used to quality assess papers in full text.

Key Definitions: Child: A child is a person 19 years or younger unless national law defines a person to be an adult at an earlier age [28].

Nutritional Status: A requirement of health of a person determined by the diet, levels of nutrients contained in the body and normal metabolic integrity. Normal nutritional status is managed by balance food consumption and normal utilization of nutrients [29].

Nutritional Deficiencies: Nutritional deficiencies, excesses, and imbalances all predispose the cell to injury. Various dietary deficiencies or imbalances of essential amino acids, fatty acids, vitamins, or minerals can lead to muscle wasting, decreased stature, increased susceptibility to infection, metabolic disturbances, and a host of other diseases, depending on which elements are missing from or disproportionate in the diet [30].

Food security: Food security exists when all people, at all times, have physical and economic access to sufficient safe and nutritious food that meets their dietary needs and food preferences for an active and healthy life. The four dimensions being: availability, access, utilization, and stability [31]. The opposite proves true for FI.

Socioeconomic status (SES): is usually measured by determining education, income, occupation, or a composite of these dimensions [32].

Immigrant: From the perspective of the country of arrival, a person who moves into a country other than that of his or her nationality or usual residence, so that the country of destination effectively becomes his or her new country of usual residence [33].

Refugee: a foreign-born individual deemed as requiring protection according to the 1951 Status of Refugee Convention who has been accepted into a host country [22]. Geopolitical events, which include war, conflicts, and climate disasters leading to displaced peoples, also encourage migrations [33].

Global migration: The drivers of migration can be conceptualized into 5 categories: economic, social, political, demographic, and environmental factors, the former 2 perceived as having the greatest effect on the volume and patterns of migration [2]. Economic motives move migration both to developed countries and to developing countries with monetary interest. Migrants seeking work but opting to remain within their regions of origin, cross to adjacent countries to satisfy their economic interests. As well as these so-called push factors, migrants are pulled for recruited labor to satiate voids in the low and high-skilled workforce [34,35,36,37,38].

Migrant flow (international): The number of international migrants arriving in a country (immigrants) or the number of international migrants departing from a country (emigrants) over the course of a specific period [33].

## 3. Results

### 3.1. Search Yields, Risk of Bias, and Quality of Reporting

The initial search identified 299 studies (Figure 1). 29 studies were included in the final analysis after 270 papers were excluded on the basis of the inclusion/exclusion criteria. A detailed description of the selected articles, with study characteristics, is reported in Table 1. The quality assessment of the full text papers included in the present systematic review is reported in Table 2.

### 3.2. Evidence Synthesis

#### 3.2.1. Overweight, Obesity, and Associated Complications

Evidence for this outcome comes from 21 papers: 13 cross-sectional studies, 5 cohort studies, 3 randomized controlled trials. Quality was determined “good” for 18 of them, and only “fair” for 3 studies. Twelve studies were conducted in the USA, the rest were performed in Canada, Brazil, China, Australia, and several countries in Europe. The studies included different variables such as SES, acculturation, ethnicity, eating habits, lifestyles including physical activity and sleeping hours, thus making the studies extremely heterogeneous.

Many studies reported a high prevalence of overweight/obesity among immigrant children.

A binational study (Mexico and California, USA) performed on 603 mother-child dyads by Rosas et al. [35] found a significantly higher prevalence of obesity among the children of Mexican descent who were born in California compared to those who were born in Mexico (53.3% vs 14.9%; *p* < 0.01). Maternal obesity was a determinant of childhood obesity in both samples, but it was significantly higher in California (49%) compared to Mexico (33%; *p* < 0.01). Although the link between maternal obesity and child’s weight status may suggest genetic or epigenetic factors, it probably also strongly reflects cultural and environmental characteristics shared with the child.

The paper of Kaiser et al. [36] examined the influence of age and gender on food patterns of Latino children in rural communities in California. 51% of the total sample was overweight or obese, with no significant effect of neither age nor gender on food patterns; mother’ s acculturation level was positively related (*p* = 0.0002) to children consumption of less healthy foods. This contradicts the findings of Rosas on the basis of quantifiable weight, although, it can be argued that certain parameters, not studied, including mental well-being, discrimination, etc. due to, or as a result of, an inability to adapt or resistance to adapt, may be looked into.

In a study by Kobel et al. [37] performed on 525 school-aged children in Germany, which analyzed the effects of interventions to promote healthier diet and behaviors, the prevalence of overweight and obesity in those with a migration background was 11.8% and obesity alone 5.7% (vs 9% and 4%, respectively in German children [38]).

Also, the study by Early et al. [39] investigated the efficacy of an intervention aimed at improving health behaviors in 68 children (mean age 10.8 years) from low-income families enrolled from 3 clinic sites in California, USA. Most of the children had a migration background with 85% being of Hispanic origin and 3% Black/African American. The reported incidence of overweight was 11.8% and that of obesity 54.4%, very similar to the figures reported by the previously cited research from the USA.

Another cross-sectional study by Geremia et al. [40] showed a high prevalence of being overweight and obese (16.3% (95% CI: 13.3–19.3) and 8.3% (95% CI: 6.1–10.5)) in schoolchildren of a southern Brazilian city with a strong Italian immigration influence. Factors such as omission of breakfast (OR 0.669 (95% CI: 0.499–0.896); *p* = 0.007), overweight and obesity in mothers (OR 1.022 (95% CI: 1.012–1.031); *p* < 0.001), age (OR 0.875 (95% CI: 0.776-0.987); *p* = 0.030) and male gender (OR 1.341 (95% CI: 1.009–1.782); *p* = 0.043) were significantly associated with excess weight.

Lane et al. [22] analyzed the health status of 300 immigrant and refugee children aged 3-13 years who had been in Canada for less than 5 years. The results indicated that older immigrant children (OR 1.18 (95% CI: 1.03–1.35); *p* = 0.017), mainly those from more privileged backgrounds, who coincidingly consumed a poorer-quality diet (OR 0.94 (95% CI: 0.90–0.99); *p* = 0.020), were at a higher risk of overweight and obesity (compared to refugees, who had a higher risk of stunting (23% vs 4.6%; *p* < 0.05)). Moreover, 52% of the newcomer children had high cholesterol levels and 29% had borderline or elevated blood pressure, higher rates compared to what reported for Canadian children (35% and 7% respectively).

Several studies analyzed the relationship between obesity and FI with conflicting results.

The study by Distel et al. [41] considered the relationship between FI and chronic stress on BMI in 104 Mexican-American children aged 6-10 years. Greater food insecurity was associated with higher BMI only when children had high levels of hair cortisol (beta=3.50, *p* = 0.02), highlighting the great influence of stress on the development of overweight and obesity.

Buscemi et al. [42] confirmed a high prevalence of overweight and obesity among Latino children in the US: those with both immigrant parents had a higher (*p* = 0.03) mean BMI percentile (85.05, SD = 18.72) compared to those with non-immigrant parents (70.30, SD = 32.84). Furthermore, the mean BMI percentile of children from food secure families was significantly higher than that of children from food insecure families (*p* = 0.022), with acculturation being a significant moderator of the relationship between food insecurity and BMI. In fact, high acculturation and high food security scores were associated with lower BMI (*p* = 0.003).

Kilanowski et al. [43] published a cross-sectional study reporting that, among migrant farm-worker families in the US, low or very low levels of food security were seen in 48% of children with under or normal weight, 75% in overweight and 53% in obese children. However, differently from Buscemi and colleagues [42], no statistical significance was found between acculturation and BMI.

The role of acculturation on the development of childhood obesity in migrant children was also explored by other authors.

In a Swiss multidisciplinary lifestyle intervention study by Ebenegger et al. [44] preschooler children of migrant parents of low education backgrounds had an increase in adiposity (*p* ≤ 0.04) and media use (*p* ≤ 0.005) and less healthy habits (more consumption of fatty foods and less of fruit (*p* ≤ 0.0001), vegetables and water (*p* ≤ 0.03)) compared to those with non-migrant or medium-high educational level parents.

The study by Huang et al. [45] assessed the association between acculturation and body weight among the children (11–17 years) of internal migrants in China. The overall prevalence of overweight/obesity was 12.5% in boys and 6.1% in girls; migrant children who had urban-to-urban migrant caregivers were more likely to be overweight/obese than those with rural-to-urban migrant caregivers (OR 1.80 (95% CI: 1.13-2.86); *p* = 0.014), and those with caregivers having higher levels of acculturation (OR 0.99 (95% CI: 0.98-1.00); *p* = 0.078) were less likely to be overweight/obese than those with lower levels. However, there was no statistically significant difference in terms of acculturation score between overweight/obese and normal weight migrant children.

The effect of the length of stay in the new country was analyzed by two papers. Using data from the National Health and Nutrition Examination Survey, a cross-sectional study by Tsujimoto et al. [46] assessed the correlation between the length of time in the US and the prevalence of obesity in 28282 children and adolescents. The prevalence of overweight/obesity was lowest in those who were foreign-born and had been in the US for less than 1 year compared to the US-born (overweight or more: 23% vs 31.8%, obesity: 8.2% vs 16.9%, severe obesity: 2.9% vs 5.9%), with an increasing trend in foreign-born children who lived in US for more than 1 year (overweight or more: adjOR 1.38 (95% CI: 0.85–2.24), obesity: adjOR 1.80 (95% CI: 0.82–3.96), severe obesity: adjOR 1.36 (95% CI: 0.25–7.30)), suggesting that foreign-born children and adolescents are susceptible to the obesogenic environment of the USA with unhealthy dietary habits (high energy, sugar, and fat intake).

In the cross-sectional study by Iguacel et al. [47] about the association between social vulnerabilities and children’ s weight status in Europe, children with a migrant background were more likely to be overweight/obese at the first evaluation (OR 1.30 (99% CI: 1.04–1.62); *p* = 0.003) and to remain so after 2 years (OR 1.14 (99% CI: 0.85–1.52); *p* = 0.239) compared to non-migrants. In this study the association between social vulnerabilities and children’ s weight was only partially explained by lifestyle factors, maternal BMI and socioeconomic status indicators. Indeed, children who accumulated more vulnerabilities (migrant background, parents unemployed, non-traditional families, lack of a social network) did not show a higher likelihood of being overweight o underweight (OR 1.33 (99% CI: 0.91–1.94)).

Other studies investigated the role of healthy behaviors (physical activity, diet, and sleep quality) in the progression towards being overweight or obese in children with migrant backgrounds.

Labree et al. [48] reported a cross-sectional study including 1943 parent–child native Dutch and immigrant dyads. Children of migrant descent had higher BMIs and prevalence of overweight and obesity (*p* < 0.05); they had lower physical activity levels and lower sleep duration, but a higher fruit and vegetable intake and a lower consumption of sugar-sweetened beverages and energy-dense snacks (all *p* < 0.05). Less sleep, low fruit intake, and more energy-dense snack consumption correlated with higher BMIs and higher prevalence of being overweight and obese (all *p* < 0.05). Ethnic differences between native Dutch and migrant children only slightly contributed to explaining differences in the prevalence of being overweight and obese.

Also, the Swedish study by Besharat Pour et al. [49] assessed the nutritional status, level of physical activity, and overweight/obesity among children of immigrant or Swedish-born parents. Children of migrants were more likely to be overweight (OR 1.33 (95% CI: 1.07–1.66)), have low physical activity (OR 1.30 (95% CI: 1.05–1.62)) and have parents with the lowest education level compared to children of Swedish parents. However, children of immigrants complied more fully with nutritional recommendations compared to the Swedish (OR = 1.31 (95% CI 1.08–1.58]), even if they had a lower intake of micronutrients such as vitamins A (mean 1049.9 ug/day, SD = 366.6) and D (mean 5.1 ug/day, SD = 1.8), calcium (mean 1162.8 mg/day, SD = 348.6) and iron (mean 10.57 mg/day, SD = 1.83), and a higher consumption of sugar and sweets (*p* < 0.01).

Chomitz et al. [50] analyzed the associations between diet, physical activity and acculturation among Chinese American children. The prevalence of overweight and obesity, as reported by the parents, was higher (32%) among children of less acculturated parents, who were 3.5 times more likely to be overweight/obese than those from most acculturated families. However, after adjustment by age, gender and income proxy, acculturation status amongst migrant children was not a significant predictor of the child health outcomes.

The cross-sectional study by Alasagheirin et al. [51] highlighted some potential risks related to body composition and metabolic features in 64 Sudanese children living in the US: 26.6% were overweight or obese determined by BMI and 28% were obese determined by body fat percent; levels of cholesterol were borderline/high in 23.4%, triglycerides in 32.8%, and 16% had risk levels for insulin resistance. Food insecurity was reported by 40% of the families; moreover, the levels of physical activity in Sudanese children were very low and 42.2% of them spent at least 3 h a day watching television or using electronics devices.

Parental feeding styles may influence children’ s eating habits and consequently their weight. This topic is highlighted by Tovar et al. [52] in a randomized controlled lifestyle intervention study that included 383 mother-child dyads of Brazilian, Haitian, and Latino peoples living in the US. 72% of mothers and 43% of the children were overweight and obese. A low demanding/high responsive feeding style in immigrant mothers (few rules, children were allowed much freedom) was significantly and positively associated with child weight status, even after adjusting for ethnicity, acculturation and stress (beta = 0.56, *p* = 0.01).

The relationship between immigrant parents’ early life deprivation and child feeding practices and weight is treated by Cheah et al. [53]. Korean and Chinese immigrant parents of children aged 3-8 years in the US were interviewed about food insecurity, acculturation, child feeding practices, evaluations whether their child weight was more or less than the ideal and child consumption of soda drink and candy; BMI was calculated for parents and children. Even if only 20% of immigrant children were overweight or obese according to BMI, parents’ FI in their childhood was associated with obesity-promoting behaviors and outcomes (beta = 0.17, *p* < 0.05).

The correlation between socioeconomic status and overweight/obesity was treated by Cook et al. [54] among seven Asian American ethnic groups. Adolescents from migrant families in the high-middle-level socioeconomic status ethnic group were less likely to be obese/overweight than those in the low-level (*p* < 0.01); this feature was not significant among US-born adolescents.

The cross-sectional analysis by Zulfiqar et al. [55] confirmed a higher trend of overweight/obesity in Australia for immigrant children from low-middle income countries (boys: 30%, 23%, 22%; girls: 35%, 22%, 24%; *p* = 0.002; for low-middle income countries, Australian-born and high-income countries respectively). The results show that the excess weight/obesity risk in this group is primarily due to sedentary activities (*p* < 0.01).

#### 3.2.2. Stunting

In agreement with WHO definitions, stunting was defined as height-for-age-and-sex less than two standard deviations below the median.

Evidence for this outcome comes from 5 papers: 4 cross-sectional studies and 1 cohort study. Quality was classified as good for all of them. The majority of the studies assessed a remarkable stunting prevalence in migrant children, highlighting the need for specific screening and therapeutic programs. Alasagheirin et al. [51] conducted a cross-sectional descriptive study of skeletal growth in 64 Sudanese children aged 5-18 living in the US and found that almost 5% (4.69%) of them were abnormally slowed in growth, and girls were nearly twice as likely to be stunted than boys (6.06% vs. 3.23%, respectively).

Similarly, the study by Lane et al. [22] documented that the prevalence of stunted growth was 4.6% in 300 migrant children aged 3-13 years who had been in Canada for less than 5 years.

In 2011 Iriart et al. [56] investigated the prevalence of stunting in Hispanic children living in the US. Among 3102 children and adolescents aged 2-19 years, the prevalence of stunting was significantly higher (*p* < 0.001) in Hispanics (6.6%) than in non-Hispanic whites (2.2%). Normal weight foreign-born Hispanic children who had been in the US 5 years or more experienced the greatest prevalence of stunting, significantly higher than their counterparts who had been in the US for less than 5 years (25.7% vs. 11.9%, *p* < 0.01). Stunting prevalence did not significantly differ between normal weight and overweight/obese Hispanic children, but overweight/obese children with economic difficulties experienced a higher prevalence of stunting, compared to those who do not face adverse socio-economic conditions (*p* < 0.10). The proportion of stunting differed significantly by country of birth (14.1% for foreign-born vs 5.2% for US-born; *p* < 0.001), language of family interview (9.4% for Spanish vs. 5.6% for English; *p* < 0.05), and number of people in the family (10.5% for 6 or more members, 3.1% for 4–5 members, 7.7% for 1-3 members; *p* < 0.001). Moreover, also low educational level (*p* < 0.10) and low family income (*p* < 0.10) were found to be associated with stunting.

Choudhary et al. [57] assessed the prevalence of stunting in children below 3 years of age migrating to Mumbai from rural and urban areas of India. This paper showed a significantly higher incidence of stunting among rural migrants compared to urban ones (*p* < 0.05) and among migrants who came to Mumbai during the last five years compared to migrants who had been staying in Mumbai for longer (*p* < 0.05), supporting a positive effect of the length of residency on the level of malnutrition due to the migrants’ capacity of adaptation into the new environment.

Evidence supporting a correlation between migration and stunting was also confirmed by Lee et al. [58], who analyzed 70 North Korean children between 6–15 years migrating to South Korea. Stunting, evaluated using the Korean Growth Chart, was significantly higher (*p* = 0.040) in North Korean children at the time of entry to South Korea (11.4%) than in South Korean children (1.0%), but interestingly, after an average length of stay of 2 years the prevalence of stunting decreased in migrating children (5.7%), but North Korean children were still shorter (*p* = 0.000), probably because linear growth deficits are slow to recover as overall nutritional status remains poor compared to peers without exposures to FI.

#### 3.2.3. Early Childhood Caries

We found one good-quality, case-control study by Werneck et al. [59] evaluating the prevalence and risk factors of early childhood caries (ECC) in Portuguese-speaking children aged 48 months or younger and with at least one parent from Portugal, Brazil, Angola, Mozambique and the Azores migrating to Canada. One-third (35%) of the children recruited had ECC. The factors that correlated the most with ECC were family without dental insurance (adjOR 4.87 (95% CI: 1.85-12.82); *p* = 0.001), lack of a family dentist (adjOR 3.96 (95% CI: 1.34–11.70); *p* = 0.013), and frequency (≥2 vs <2) of snack consumption (adjOR 3.79 (95% CI: 1.32-10.83); *p* = 0.013). Subjects from low-income families were almost 4 times more likely to have ECC, and low parental knowledge of harmful feeding habits were linked to higher prevalence of ECC (OR 2.84 (95% CI: 1.27–6.33); *p* = 0.010). Also, parents’ countries of origin and their age at the time of immigration were significantly associated with ECC: children of parents from non-European countries migrating to Canada had a higher occurrence of ECC (*p* < 0.026), and parents who migrated in their 20s or older were 2-4 times more likely to have a child with ECC than those who immigrated at younger age (*p* < 0.04). A person who immigrates when he or she is a child would be more exposed to positive oral health messages at school, and this is not the case for someone who immigrated as an adult. Surprisingly, oral hygiene history was not a factor influencing dental health.

#### 3.2.4. Micronutrient Deficiency

Evidence for this outcome comes from 3 cross-sectional and 1 cohort studies. Quality was classified as good for 3 of them and fair for one. These studies reported a high prevalence of micronutrient deficiency in migrant children.

Vitamin D deficiency was evaluated by Vatanparast et al. [60], distinguishing between migrant and refugee children aged 6-11 years in Canada. They noted a mean serum 25 hydroxyvitamin D (25(OH)D) concentration significantly higher in non-immigrant children than in immigrant or refugees, with overall a vitamin D deficiency/inadequacy in 63% of migrants and 80% of refugees. The prevalence of calcium intake inadequacy was 74% in migrants and 77% in refugees, and of vitamin D intake inadequacy 81% in migrants and 97% in refugees. Female sex (Standardized Regression coefficient: −0.39 ± 0.11, *p* < 0.001), longer stay in Canada (Standardized Regression coefficient: −0.23 ± 0.11, *p* = 0.04), darker skin pigmentation (Standardized Regression coefficient: 0.24 ± 0.11, *p* = 0.027) and low vitamin D intake from food and supplement (Standardized Regression coefficient: 0.46 ± 0.11, *p* < 0.001) were found to be significantly associated with deficient vitamin D serum status. Both immigrant and refugee children had percentile height (mean ± SD) lower than the 50th. Height and serum vitamin D were found to be significantly related to the Total Body Bone Mineral Content (Standardized Regression coefficient: 0.13 ± 0.06, *p* = 0.047), which was low in 41.9% of migrant children and 35.9% of refugee children compared to threshold for age, sex, and ethnicity.

Iron status, using the storage protein ferritin as a proxy, was assessed by Sacri et al. [61], who performed a cross-sectional hospital-based study in France on 657 children from 6 months to 6 years with no signs of active inflammation (CRP < 10 mg/L), because it might influence serum ferritin status. 14% had a migrant mother. Having a migrant (95% CI: 25.9–34.0) or unemployed (95% CI: 33.3–44.6) mother was significantly (*p* < 0.05) related to low serum ferritin (32 vs 45 µg/L for migrant vs non-migrant mother and 37 vs 50 µg/L for unemployed vs employed mother). Iron deficiency (ID) prevalence was 2.8% to 3.2% depending on serum ferritin threshold of 10 and 12 µg/L. Male gender (OR 2.17 (95% CI: 1–4.70)), mother being a migrant (OR 3.12 (95% CI: 1.06–9.18)), underprivileged family status (OR 3.83 (95% CI: 1.39–10.53)) and low maternal education (OR 4.48 (95% CI: 1.26–15.98)) were related to ID.

However, Saunders et al. [62] found no association between immigrant status and serum ferritin, iron deficiency and iron deficiency anemia in 2614 healthy children aged 12–72 months living in Toronto, 47.6% of whom had an immigrant family status. The median serum ferritin was 30 µg/L (IQR 19–39); 10.4% of the children had ID, and 1.9% had ID anemia. Older age (*p* < 0.001), male sex (*p* = 0.02), no or low cow’s milk intake (*p* < 0.001), and shorter breastfeeding duration (*p* < 0.001) were significantly related to higher serum ferritin, while immigrant status was not related to lower levels of serum ferritin. Interestingly, a household income of £15 000–£29 999 was associated with a 24.1% increase in child serum ferritin compared to children from families with a higher household income (*p* = 0.002).

Kim et al. [63] analyzed dietary intakes and plasma concentrations of vitamin E, vitamin C, selenium and carotenoid in 29 non-supplemented Latino children aged between 4–8 years migrating to Nebraska, USA. They reported that the mean daily intake of vitamin E was lower than the Estimated Average Requirement (EAR) and Recommended Dietary Allowance (RDA), but none of the subjects consumed less than the RDA for vitamin C (25 mg/day), selenium (30 µg/day), and carotenoid according to the age. Vitamin E plasma concentration was insufficient in 31% to 69% of the subjects depending on the criterion used, while mean vitamin C, selenium, and carotenoid concentrations were indicative of adequacy.

## 4. Discussion

This systematic review assessed the association between migration, FI, and major diet-related morbidities in children using data from 28 studies, 26 of whom were observational (cross-sectional, and cohort) studies, while the two others were randomized controlled trials. All but 3 studies were classified as being of good quality (Table 1).

Overweight and obesity were the most investigated problems, but the results were sometimes conflicting, probably because of the different variables (different populations, SES, acculturation, lifestyles…) that were taken into account by the single works. Most of the studies highlighted a higher prevalence of overweight/obesity in migrant than in non-migrant children [35,37,39,40,42,47,48,49]. Obesity is linked to difficulties in social adjustment, low self-esteem, depression, and lower academic achievement [64,65], causing a significant reduction in the children’s quality of life [66]). Moreover, overweight children are at a greater risk for remaining overweight in adulthood and for the development of chronic illnesses, such as diabetes and cardiovascular diseases [67], which increases premature mortality [68]. Indeed, high cholesterol levels, high triglycerides, and high blood pressure have been observed frequently in migrant children [22,51].

The increased risk of being overweight in children with a migration background is affected by multiple factors, such as maternal obesity [35,40], given that women commonly assume the primary responsibility for the care and feeding of children using familiar lifestyle behaviors [69], lower economic income [50,54,55], and male gender [40]. Indeed, boys, even if generally more active than girls, have been reported to consume fewer vegetables and more sweet drinks compared to girls of the same age and to watch television more hours per day [70], thus unbalancing the ratio between caloric intake and consumption. A lower level of acculturation has been found to be associated with a higher BMI [42,45,50], mainly if both parents were of low educational level [44]. This could be explained, according to other studies, by the fact that unhealthy eating behaviors and sedentary habits are more often observed in children from low educational level families [71,72]. Prevalence of overweight/obesity has been found to be lower in foreign-born children and in children living in the host country for less than 1 year [46,55]. This could be attributed to immigrant self-selection, meaning that the ones who migrate are the healthiest subjects who are able to endure a long and physically demanding journey and who, at the arrival in the host country, are in reasonably good health (the so-called “healthy immigrant effect”) [73]. Interestingly, a convergence of migrant health levels to native health ones within approximately 10–20 years has been reported. [73] Moreover, we should consider that BMI is frequently low at the arrival, mostly because in the countries of origin there is not food over-consumption and, in particular, low consumption of fatty, sugary and processed food, that is instead high in the destination countries (although it is true that obesity is increasing worldwide and it is high in some middle-income countries such as Mexico) [74]. The increase in the prevalence of overweight and obesity over time in the host country among migrant children might be due to acquired environmental risk factors (i.e., western-diet, availability of low-cost high-sugar and high-caloric snacks and a sedentary lifestyle) combined with those from the country of origin (i.e., restricting physical activity [49,51], and cultural preferences for large body size as a proxy for wealth, health, or beauty) [75,76]. Furthermore, parent’s early life deprivation in the countries of origin, where FI and under-nutrition could be significant and dangerous factors to impaired childhood health, has been reported to be associated with parenting behaviors that promote weight gain [77], due to their different perception of what a healthy child weight is [53].

FI appeared to be discordantly correlated to overweight. It could be hypothesized that a higher BMI is easily associated with FI [41,43], given the availability of low-cost energy-dense foods; members of food insecure families report overeating, when possible, foods they like and even dislike to compensate for periods of deficiency [78]. This link could also be explained by the limited knowledge, time and resources that low food-secure subjects experience to engage in healthful eating and physical exercise [79]. It is interesting that the FI-obesity link has been reported to be stronger among women than among children: mothers frequently sacrifice their food supply to ensure that children are food secure, and this is supported by the fact that only half of food insecurity families with children are also child food-insecure. [9]. In contrast, some authors noticed that more food secure migrant families have children with higher BMI than the food insecure ones [38]. Families who can afford to purchase high-cost food such as fruits and vegetables, could also afford to buy more food in general and consequently cause a major intake of calories in children [80].

We also found strong evidence of a link between migration and stunting [22,51,56,57,58]. Stunting may reflect the cumulative effect of chronic malnutrition and not only under-nutrition [56] and it is associated with increased morbidity and mortality from infection, in particular, pneumonia and diarrhea [81], and short and long-term functional impairment, including poor cognition and educational performance in childhood, low productivity and low wages in adulthood [82]. Iriart and colleagues [56] showed a higher prevalence of stunting in overweight children living in families with economic difficulties compared to overweight children who do not experience such conditions, suggesting that being overweight often matches with a high-calorie but nutrient-poor diet. A shorter period of stay in the host country, a higher number of people in the family, being foreign-born, a low educational level, and socio-economic difficulties affect the prevalence of stunting [56,57,58]. As the years of stay increase, migrants incorporate to the new environment and progressively learn new habits, beliefs, and behaviors, [57,58] and this effect is even more valid in children born in the host country from migrant parents [56]. Probably, by being incorporated into the new environment over time, immigrants learn how to get better food and progressively improve their SES.

Concerning oral health, ECC was found to be more frequent in migrant children from low-income families, and with parents who migrate from non-European countries in their 20s or later. If untreated, ECC evolves to a more severe disease, which can lead to malocclusions, abscesses and pain [21] and can negatively impact on quality of life, nutritional status, and growth [83,84]. It has been also reported that dental problems, left untreated, can accelerate diabetes and cardiovascular disease complications and may be also associated with respiratory diseases and adverse pregnancy outcomes [85].

When migrants arrive in the host country, they face a dental health care system that is predominantly private [86] and most of them cannot access this care due to a lack of financial resources and because they are not covered by dental insurance [87]. Indeed, we found that the other factors that strongly correlated with ECC were family without dental insurance and the absence of a family dentist [59].

A lack of a balanced diet is the cause of micronutrient deficiency, most of all vitamin D deficiency and ID. Serum vitamin D concentration has been noted to be low in a high rate of migrants, due to an inadequacy of vitamin D and calcium intake, and this was significantly connected to low Total Body Mineral Content. Other risk factors include being of female gender and having dark skin pigmentation [60]. Women are usually at a greater risk than men due to religious practices stemming from various parts of the word, causing insufficient skin exposure to the sun. Furthermore, dark skin pigmentation reduces the quantity of vitamin D produced for a given amount of solar ultraviolet radiation, that is typically lower in high-income countries than in the countries of origin of most migrant people; moreover, light exposure in the host countries is often insufficient during wintertime. Inadequate levels of vitamin D prevent proper bone growth, which can lead to rickets in children [15]. Clinical manifestations of rickets include restlessness, irritability, frontal bossing, wide open fontanelles, skeletal signs (enlargement of costochondral junctions at 6–9 months of age), soft osseous border, teething delayed, profuse sweating, muscle flabby, and upper respiratory tract infections [88]. Skeletal consequences include stunting, developmental motor delay, and deformities. In severe cases, the development of hypocalcemia can lead to tetany, seizures, laryngospasm, cardiomyopathy, and even death [89]. In order to prevent serious disease, consumption of milk and alternatives, vitamin D-fortified food, and vitamin D supplements should be recommended, in addition to regular sun exposure which is also associated with less sedentary lifestyles.

Two studies focused primarily on ID. ID is strictly associated with an inappropriate diet [14], and severe and/or prolonged ID can evolve to anemia. Notably, preschool age children are at a higher risk because of their rapid phase of growth, leading to a rapid depletion of the iron stores when the intake is not sufficient. The occurrence of ID anemia is also adversely affected by intestinal parasitosis [90], whose prevalence has been reported to be around 20% in newly arrived migrants in an Italian study [91]. We found conflicting results that correlate ID to migration. Strong risk factors raised by these studies are low socio-economic status, high cow’s milk intake which is frequently used because infant formulas are more expensive, and longer breastfeeding duration. Iron is an indispensable micronutrient in brain metabolism, and its deficiency can cause changes in the neurotransmitter homeostasis, decrease myelin production, decline synaptogenesis, and decline the function of the basal ganglia, which adversely affect the psychomotor development and mental capacity [92]. ID can also affect children’s emotional and psychological behavior, and this is correlated with the modification in the metabolism of dopamine, GABA, hippocampal function and structure, and myelinization [93]. ID anemia is also implicated in the susceptibility to infections, mainly of the upper respiratory tract, which are more frequent and have longer duration in anemic than in healthy children [94]. Early identification of ID is important for preventing the potentially irreversible effects of long-term low iron stores, such as developmental delay, mostly in the fields of attention, motivation [95], stimulus encoding, and memory [96]. Preventative strategies involving parents’ education about the consumption of naturally iron-rich and of iron-fortified foods since breastmilk weaning could be implemented, as well as the routine exclusion of parasitic infections that can impair iron intestinal absorption and contribute to ID anemia [97].

Most of the studies were performed in the USA, and not much is known about dietary-related problems in children migrating to other areas of the world. The link between overweight/obesity and migration to the USA, mainly in the Latino people, was reported by several authors. However, studies about diet-related morbidities and migration to other countries are scarce and diverse, so that no clear association could be found between specific diseases and migration to specific geographical areas other than Northern America. Moreover, the diverse data which were variably considered in the papers (SES, acculturation, various ethnicities, eating habits, lifestyles) made the studies very heterogeneous between each other and this might account for the, sometimes, conflicting results.

Unhealthy eating habits can undermine healthcare systems. Obesity starting from infancy will probably continue throughout adulthood, and lead to chronic diseases such as diabetes, hypertension, and cardiovascular diseases, which have a strong impact on our healthcare systems [98]. There is solid evidence that programs of screening and interventions could be effective to improve the health of migrant children [99,100]. Many pediatric scientific medical societies actively recommend universal screening for social determinants of health, including FI, in order to identify people at risk at a social level. Unfortunately, although tools to efficiently assess social determinants exist, they are still lacking comprehensiveness in providing a concise view of health in socially deprived populations [101,102,103,104,105]. Therefore, public health programs to evaluate the increased frequency of micronutrient deficiencies and diseases in these populations need to be promoted.

## 5. Conclusions

The present systematic review identified a clear association between migration and diet-related morbidities in children, the major risks being obesity and stunting due to chronic malnutrition. FI and frequently low levels of acculturation are important social determinants in migrant people, leading to the purchase of low-cost foods with poor nutritional value and the maintenance of bad food habits and incorrect lifestyles. However, most of the studies were performed in the USA, and not much is known about dietary-related problems in children migrating to other areas of the world; more work should be done to provide clear evidence for other potentially important morbidities in this group.

Health policies should take into account these areas of vulnerability, which might be addressed with specific screening and educational programs. Healthy policies play a key role to face childhood obesity and the related disease, but currently, they are still at an early stage and have no perceptible impact on this problem [106,107]. Governments need to implement regulations that make a healthy diet accessible and affordable for all, especially for the most disadvantaged populations at major risk of FI; such changes sound necessary to improve chronic disease prevention and control, from the perspective of the economic sustainability of healthcare [108]. Paying for primary prevention, mostly in childhood, often leads to an overall saving: “Pay now or pay (more) later” [109]. In the same way, integrating oral health into primary care could help increase access to preventive care, treatment, and promote overall health, for example implementing dental hygienists into primary care settings and educating primary care clinicians in the impact of oral diseases, trains them in risk assessment, and how to assimilate oral health preventive services into their practice [110].

Patient-centered care and engagement have been reported to lead to greater patient satisfaction, improved clinical outcomes, health service efficiency, and improved health-related business metrics [111]. As Geoffrey Rose stated at the end of the past century, “the primary determinants of disease are mainly economic and social, and, therefore, its remedies must also be economic and social. Medicine and politics cannot and should not be kept apart.” [112].

## Figures and Tables

**Figure 1 nutrients-12-00379-f001:**
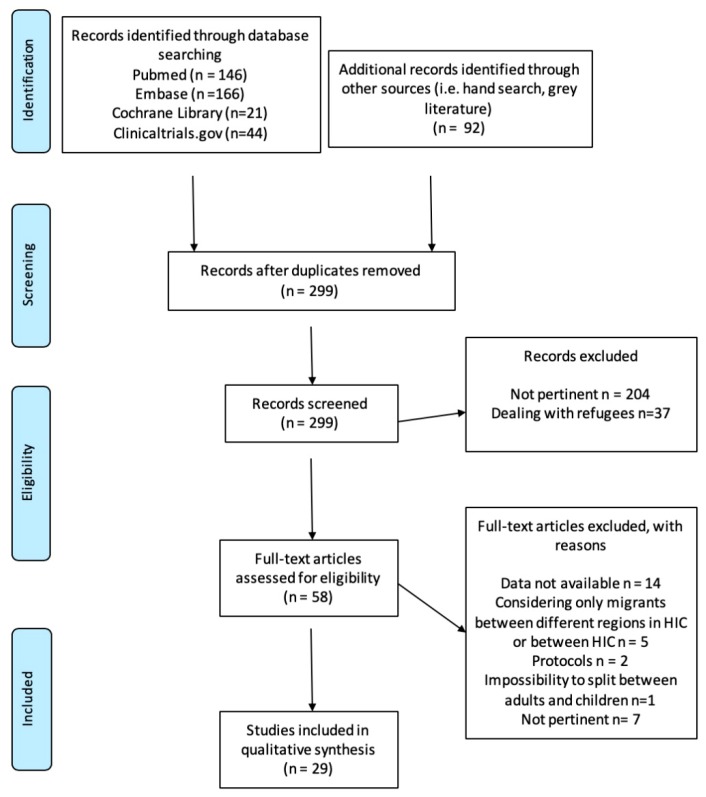
Preferred reporting items for systematic reviews and meta-analyses (PRISMA) 2009 Flow Diagram showing the process for articles selection [26]. HIC= high income countries.

**Table 1 nutrients-12-00379-t001:** Characteristics and main results of the included studies. [27].

Author, Year	Migrating from	Immigrating to	Study Design	N, Migrant Characteristics	Age	Outcome Disease	Dietary/Cultural Issues	Main Results	QA
Rosas, 2010 [35]	Mexico	USA (California)	cross-sectional	603 mother-child pairs Mexico-born and California born-children of Mexican descent	5 year	overweight/obesity	Maternal obesity was a determinant of childhood obesity in both settings; in Mexico, male gender, high SES and low food security were determinants of childhood obesity.	prevalence of childhood obesity is much higher among children of Mexican descent	13
Kaiser, 2015 [36]	Mexico	USA (rural communities in California)	cohort	217 children of Latino descent	2–8 year	overweight/obesity	Preference of american-style foods after children enter the public school system	51% of children overweight/obese	14
Kobel, 2017 [37]	anywhere (mostly Turkey and Russia)	Germany	randomized controlled trial	525 children with migration background	7.1 ± 0.7 year	obesity	Lower fruit and vegetable consumption, higher screen media time, lower physical activity	children with migrant background: 11.8% overweight/obese (vs 9% non migrant) – 5.7% obesity if migration background	14
Early 2019 [39]	Hispanic (85%), Black/African (3%)	USA (California)	quasi-randomized controlled trial	68 children	10.8 year	overweight/obesity	inadequate fruit and vegetable consumption, excess of sugar-sweetened beverages	overweight prevalence: 11.8%; obesity prevalence: 54.4%	12
Geremia, 2015 [40]	Italy	Brazil	cross-sectional	590 Italian immigrant children	9–18 year	overweight/obesity	Low frequency of consumption of vegetables, more fat foods.	1) High prevalence of overweight and obesity in this city; 2) factors such as omission of breakfast, overweight and obesity in the mother, age and male gender were associated with excess weight.	12
Lane, 2018 [22]	Asia, Middle East, Africa, Latin America, Europe o US.	Canada	cross-sectional	300 immigrant and refugee children	3–13 year	health status (stunting, overweight/obesity, hypertension, cholesterol levels, health disparities)	Dietary changes (western diet); many immigrants and refugees vulnerable to health disparities	1) Refugee children are at risk of stunting while immigrants are more at risk of overweight/obesity, especially if they are older and they are from privileged backgrounds in low-income countries; 2) 29% of newcomer children had borderline or elevated blood pressure and 52% high cholesterol levels	14
Distel, 2019 [41]	Mexico	USA	cohort	104 Mexican American children	8.39 year (6-10)	obesity	Food insecurity and chronic stress	Greater food insecurity associated with higher BMI only when children had high levels of hair cortisol	16
Buscemi, 2011 [42]	Latino	USA	cross-sectional	63 Latino children	2–17 year	obesity	Acculturation as a moderator of the relationship between food insecurity and BMI: higher acculturation and high food security associated with lower BMI	Mean BMI percentile significantly higher for immigrants (85) than non immigrants (70); mean BMI percentile 91 for food secure families and 71 for food insecure (s.s.)	14
Kilanowski, 2012 [43]	Latino	USA	cross-sectional	60 parent–child dyads of migrant farmworkers	2–13 year	overweight/obesity	55% low or very low household food security; surprisingly, children of migrants better than peers concerning fruit and vegetable consumption	22% overweight, 26% obese; low or very low levels of food security were seen in 48% of children under-normalweight, 75% overweight, 53% obese	14
Ebenegger, 2011 [44]	Portugal, Albania/Kosovo, other European countries; Africa, Asia, Latin America, other	Switzerland	cross-sectional	542 children of migrant (71%) and non-migrant parents	5.1 ± 0.6 year	overweight/obesity	Children of migrant and low education level parents ate more meals and snacks while watching TV, more fatty foods and less fruit	Children of migrant parents had higher weight, BMI and % body fat compared to non-migrant; parental migrant status and educational level independently contributed to adiposity and eating habits	14
Huang, 2018 [45]	underdeveloped area in China	developed area in China	cross-sectional	1154, children-caregiver dyads, internal migrants	11–17 year	overweight/obesity	Levels of acculturation negatively associated with overweight/obesity; children with urban-to-urban migrant caregivers more likely to be overweight/obese than those with rural-to-urban migrant caregivers	9,7% overweight/obese (> males, >11-13 yrs rather than 14-17 yrs, >urban-to-urban rather than rural-to-urban)	15
Tsujimoto, 2016 [46]	Mexico, non-hispanic white, non-hispanic black	USA (Boston)	cross-sectional	28282 foreign-born and US-born children	2–19 year	overweight/obesity	Obesogenic environment	Prevalences of overweight/obesity lowest in children/adolescents foreign-born and who had been in the US for <1 yr, highest in the US-born (overweight: 23% vs 31.8%, obesity: 8.2% vs16.9%, severe obesity: 2.9% vs 5.4%). Risk of being overweight/obese for US-born vs foreing-born in the US for <1 yr: aOR 2.2 overweight, aOR 3.15 obesity.	12
Iguacel, 2018 [47]	anywhere	Belgium, Cyprus, Estonia, Germany, Hungary, Italy, Spain, Sweden	cohort	8624 children of migrant (13,4%) and non-migrant families	2–9.9 year at baseline 4–11 year after 2 years	overweight/obesity	Partially explained by lifestyle factors (mainly sedentary habits e.g., screen time)	Overweight/obesity at T0: 23.4% migrant origin vs 16.8% native, at T1: 28.5% migrant origin vs 21.5% native (OR 1.3); children with migrant background were more likely to remain o/o after 2 yrs compared to non migrants (OR 1.29)	15
Labree, 2015 [48]	Turkey, Morocco, other western and not-western countries	Holland (Rotterdam and Eindhoven)	cross-sectional	1943 immigrant and native parent–child dyads	8–9 year	overweight/obesity	Low sleep duration, low fruit and high snack intake associated with higher BMIs and prevalence of overweight/obesity; ethnic differences in sleep duration and dietary intake did not have a large impact on ethnic differences in overweight/obesity	children of migrants had ss higher BMI and higher prevalence of ovw/ob, lower prevalence of underweight; higher intake of fruit and vegetables and lower intake of snacks and sweet drinks; lower sleep duration. Less sleep, low fruit intake, and more energy-dense snack consumption correlated with higher BMIs and higher prevalence of overweight and obesity	13
Besharat Pour, 2014 [49]	anywhere (Africa, Asia, LatinAmerica, Europe excluding Sweden, and Sweden)	Sweden	cohort	2589 immigrant (22%) and non-immigrant children	8 year	obesity	Immigrants: higher consumption of fruit/vegetables but also of cakes and sweet	Being overweight and having low physical activity more common among children of immigrant parents (>both immigrant parents)	13
Chomitz, 2017 [50]	Asian (>Chinese)	USA (Chinatown in Boston)	cross-sectional	132 Asian American children	4.9 year (3.5–6)	obesity	Perceived parents’ barriers: 1) worry about safety when child plays outside 2) healthy food too expensive	32.6% overweight/obese (vs 23.4% overall USA and 9% Asians) - children of less acculturated parents, more likely with lower income or recent immigrants, were 3,5 times more likely to be overweight/obese than those of more acculturated parents - more acculturated more likely to provide fruits daily or more but also more sugary snacks	14
Alasagheirin, 2018 [51]	Sudan	USA	cross-sectional	64 immigrant and refugee Sudanese children	5–18 year	growth, body composition, metabolic risk, physical activity and food security	Food insecurity in 40% of families, sedentary habits reported by many	32% obese, 46% low lean mass; high cholesterol 23%, high triglycerides 32%, high insulin resistance 15% (correlates with high risk of diabetes and cardiovascular problem); Low bone mass which could contribute to osteoporosis	13
Tovar, 2012 [52]	Haiti, Latin America, Brazil	USA	randomized controlled trial	383 mother-child dyads	mothers 20–55 year, children 3–12 year	overweight/obesity	A low demanding/high responsive feeding style is significantly and positively associated with higher child weight.	72% of mothers and 43% of children overweight/obese. Fifteen percent of mothers reported their feeding style as being high demanding/high responsive; 32% as being high demanding/low responsive; 34% as being low demanding/high responsive and 18% as being low demanding/low responsive.	18
Cheah, 2012 [53]	China, Korea	USA	cross-sectional	130 children of first-generation immigrants from China (62%) and Korea (38%)	3–8 year	obesity	Parents’ food insecurity in their childhood associated with obesity-promoting behaviors and outcomes	20% overweight/obese; parents with food insecurity in childhood: 1) evaluated their children weighing less than ideal 2) allowed more servings of soda and sweets; early life material deprivation: 1) less concern about children’s diets, 2) less concern child eating too much or becoming overweight	14
Cook, 2017 [54]	Asia	USA (California)	cohort	1525 Asian American adolescents	12–17 year	overweight/obesity	The two lifestyle factors (i.e., physical activity and fast food consumption) were not associated with overweight/obesity	Overweight/obesity rate higher among those with lower (24.7%) than higher (13.4%) family incomes; higher among those in the low-SES (29.0%) than middle/high-level SES ethnic groups (11.6% and 12.8%, respectively). By ethnicity, overweight/obesity lowest among Japanese (4.8%) and highest among Filipino (26.3%) and Southeast Asians (25.5%). Adolescents in high-middle SES were far less likely to be overweight or obese than those in low SES: this was more pronounced for foreign-born adolescents vs US-born.	14
Zulfiqar, 2018 [55]	HIC and LMIC	Australia	cross-sectional	4115 children originating from Australia (58%), HIC (30%), LMIC (12%)	4–11 year	overweight/obesity	Higher intake of both vegetables and sugar-sweetened-beverages, higher inclination toward sedentary activities and lower organized sports participation.	Higher overweight/obesity rates in immigrants from LMIC (LMIC/HIC/Australian %: boys 30/23/22, girls 35/22/24) and higher in second-generation	15
Iriart, 2011 [56]	Hispanic	USA	cross-sectional	3102 Hispanic (38%) and non-Hispanic children	2–19 year	stunting	Hispanics more likely to be less than full food secure (30.5% vs 11.8%); hispanics with normal weight were more likely to be fully food secure	Hispanics: highest proportion of stunting (6.6%), overweight/obesity (39.3%), stunting among normal weight (7%), stunting among overweight/obese (6%) compared to non-hispanic whites, non-hispanic blacks, other races; tendency for overweight/obesity in hispanic children who face adverse socioeconomic conditions to experience a higher prevalence of stunting	15
Choudhary, 2009 [57]	44% rural, 51% urban	Mumbai city	cross-sectional	481 internal migrant children	<3 year	stunting, underweight, wasting	poverty	Stunting and low mother-BMI >in migrants with high disadvantage for rural migrants; as the years in Mumbai increase, migrants learn to assimilate to the new environment and the disadvantage compared to non-migrants declines	15
Lee, 2015 [58]	North Korea	South Korea	cohort	70 immigrant children	6–15 year	stunting and obesity	Those who lived in South Korea longer were less likely to be currently stunted	At entry 11.4% stunted and only 5.7% after 2 years. The prevalence of obesity was similar to that of SK children. The likelihood of remaining stunted was significantly associated with older age and shorter residency in SK. The was no significant association with food security situation at birth.	14
Werneck, 2008 [59]	Portugal, Brazil, Angola, Mozambique, Azores	Canada (Toronto)	case-control	104 immigrant children	≤48 months	Early childhood caries	Immigrants have difficulty in obtaining dental care primarily; factors that correlated the most with early childhood caries were family without dental insurance, lack of a family dentist, and frequency of snack consumption.	35% early childhood caries	15
Vatanparast, 2013 [60]	Asia, Africa, Middle East and Latin America	Canada	cross-sectional	72 children (33 immigrant and 39 refugee)	6–11 year	vitamin D deficiency	Calcium intake inadequacy 74% in migrants and 77% in refugees, vitamin D intake inadequacy 81% in migrants and 97% in refugees	Serum vitamin D deficiency/inadequacy in 63% of migrants and 80% of refugees	12
Sacri, 2017 [61]	anywhere	France	cross-sectional	657 immigrant (14%) and non-immigrant (86%) children	3.9 year (<6)	iron deficiency	Male gender, mother being a migrant, underprivileged family status and low maternal education were related to ID	Significantly associated with mother being a migrant: low serum ferritinemia (32.1 mcg/L vs 44.9) and iron deficiency (7% vs 2%); iron deficiency prevalence 2.8% to 3.2% depending on serum ferritin threshold of 10 or 12 µg/L.	15
Saunders, 2016 [62]	from industrialized and not industrialized countries (especially from Europe)	Canada	cross-sectional	2614 children (47.6% immigrant)	12–72 month	iron deficiency	Younger age, male sex, high cow’s milk intake, longer breastfeeding duration associated with lower serum ferritin	No association between family immigrant status and iron status, no need for iron supplementation in immigrants’ children	15
Kim, 2006 [63]	Latin America	USA (Rural Nebraska)	cohort	29 Latino immigrant children	4–8 year	micronutrients deficiency (plasma concentrations of vitamin E, vitamin C, selenium and carotenoids)	59% reported consuming less than the Estimated Average Requirement for vitamin E; in general, consumption of the Recommended Dietary Allowances for vitamin C and selenium	69% vitamin E inadequacy; in general, normal levels of vitamin C and selenium.	13

SES: socioeconomic status; HIC: high-income countries; LMIC: low-middle-income countries; QA: quality assessment according to the EPHPP.

**Table 2 nutrients-12-00379-t002:** Quality assessment of the included studies according to the EPHPP [27].

Author, Year	Study Design	Blinding	Selection Bias	Data Collection	Confounding	Withdrawal and Drop-Outs	Overall Rating
Rosas 2010 [35]	1	2	3	3	3	1	13
Kaiser 2015 [36]	1	2	3	3	3	2	14
Kobel 2017 [37]	2	2	3	3	2	2	14
Early 2019 [39]	1	2	3	3	2	1	12
Geremia 2015 [40]	1	2	2	3	2	2	12
Lane 2018 [22]	1	2	3	3	3	2?	14
Distel 2019 [41]	2	2	3	3	3	3	16
Buscemi 2011 [42]	1	2	3	3	3	2	14
Kilanowski 2012 [43]	1	2	3	3	3	2	14
Ebenegger 2011 [44]	1	2	3	3	3	2	14
Huang 2018 [45]	1	2	3	3	3	3	15
Tsujimoto 2016 [46]	1	2	3	3	2	1	12
Iguacel 2018 [47]	1	2	3	3	3	3	15
Labree 2015 [48]	1	2	2	3	2	3	13
Besharat Pour 2014 [49]	1	2	3	3	2	2	13
Chomitz 2017 [50]	1	2	3	3	2	3	14
Alasagheirin 2018 [51]	1	2	3	3	2	3	13
Tovar 2012 [52]	3	3	3	3	3	3	18
Cheah 2012 [53]	1	2	3	3	3	2	14
Cook 2017 [54]	1	2	3	3	3	2	14
Zulfiqar 2018 [55]	2	2	3	3	2	3	15
Iriart 2011 [56]	1	2	3	3	3	3	15
Choudhary 2009 [57]	2	2	3	3	2	3	15
Lee 2015 [58]	2	2	3	3	2	2	14
Werneck 2008 [59]	2	3	3	3	3	1	15
Vatanparast 2013 [60]	1	2	3	3	2	1	12
Sacri 2017 [61]	1	2	3	3	3	3	15
Saunders 2016 [62]	1	2	3	3	3	3	15
Kim 2006 [63]	1	2	3	3	2	2	13

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
