# Peer review of "Food Insecurity and Major Diet-Related Morbidities in Migrating Children: A Systematic Review"

_nutrients, 2020, doi:10.3390/nu12020379_

Round 1
Reviewer 1 Report
The authors present an interesting review in a hot topic: nutritional status in immigrant families and health outcomes. The research questions are clearly set. There are some minor questions that raise after reading the paper:
Comments
Abstract The authors should avoid personal interpretation on the subject and provide with objective data coming form the analysis of the studies. See: “We strongly believe…” to the end of the paragraph should be rewritten in that way Introduction The references for FI in immigrant families come only from US (10,11). Are there data form other scenarios, mainly Europe?. If not, remark on the text. Material and methods It is obvious that reports published after Nov 2019 are not included, do the authors need to include this condition as an exclusion criteria? EPHPP (line 111) should be written in the complete form Figure 1. Ad as a footnote the meaning of HIC should be explained Table 1. Ref 41 refers to children moving within the same country. It does not fit within the scope of the review. The same for ref 53 (urban vs rural within the same country) The authors have included in the review some intervention studies that are clearly different from those observational studies: ref 33, 35, 48
Table 2 should fit better as supplementary information than as a table
Author Response
We thank the reviewer for appreciating our paper. Here is a point-by-point reply to their comments.
- Abstract The authors should avoid personal interpretation on the subject and provide with objective data coming form the analysis of the studies. See: “We strongly believe…” to the end of the paragraph should be rewritten in that way
Reply: Thank you for your comment; the abstract was modified according to your suggestion (ll.31-32)
- Introduction The references for FI in immigrant families come only from US (10,11). Are there data form other scenarios, mainly Europe?. If not, remark on the text.
Reply: To the best of our knowledge, only few papers addressed the issue of FI in European children and they did not analyze the role of migration. This point was added to the introduction (ll.64-68)
- Material and methods It is obvious that reports published after Nov 2019 are not included, do the authors need to include this condition as an exclusion criteria?
Reply: You are right, that was redundant; the date of publication was removed from exclusion criteria
- EPHPP (line 111) should be written in the complete form
Reply: Done (l.149)
- Figure 1. Ad as a footnote the meaning of HIC should be explained
Reply: Done (l.208-209)
- Table 1. Ref 41 refers to children moving within the same country. It does not fit within the scope of the review. The same for ref 53 (urban vs rural within the same country)
Reply: Thanks for this comment, which allows us to better specify the concept of our inclusion. We decided to take into account even those studies, in fact, since our aim was to evaluate differences in migrants going from low to high-resource settings, we did not assume that the concept of migrant was only between countries but could also be between regions. Speaking, in particular, of these two papers, they described migration between different areas of very big and heterogeneous countries like China and India are. In any case, we specified in the inclusion criteria that migration could be also considered between different regions in the same country, given that it was from low to high-resource settings (ll. 131-132).
- The authors have included in the review some intervention studies that are clearly different from those observational studies: ref 33, 35, 48
Reply: Thank you for the comment. We decided to take into account all the types of study design (including RCTs, like the ones you have highlighted) in order to be the most comprehensive as possible. Given the fact that we did not carry out any metanalysis due to the high heterogeneity of the papers, we think that this was the best trade-off in order to analyze the vast majority of the literature on this topic. Systematic review methods have been subject to considerable discussion and debate especially regarding the selection of studies to include or exclude from review. The traditional approach believes that they must deal with all studies regardless of the quality. The critical evaluation approach aims to include only studies that meet a predetermined threshold of quality. Adherence to either approach may impose serious limitations. As an alternative, some systematic reviewers advocate an intermediate approach that considers the merits of both positions (Slavin, R. E. (1987). Best-evidence synthesis: Why less is more. Educational Researcher,16,15–16).
- Table 2 should fit better as supplementary information than as a table
Reply: Thank you for this comment; we agree with your opinion and moved the table to the supplementary material
Reviewer 2 Report
This is a systematic review evaluating the food insecurity and nutritional status of vulnerable children during migration. This is an interesting question and one that would have been good to have in the nutrition community. Unfortunately a significant methodological flaw makes the paper unacceptable in my mind for publication in its current form. The authors could consider repeating the literature search with more proper attention to inclusion and exclusion criteria, etc. A systematic review that is not done systematically and rigorously is really of little use to the field.
The inclusion of migrants but the exclusion of refugees is subtle when considering issues of food insecurity and malnutrition. I can understand the motivation and rationale for keeping the two populations separate, as they have different risk factors and outcomes, but this difference may be lost on casual readers. I would recommend that in the Abstract and Introduction, this should be made much more explicitly clear and emphasized, including an explanation of the rationale and examples of how these two populations may reasonably assumed to have differing risk factors and outcomes.
Given that refugees are to be excluded in this review, then why was “refugee*” included in the search string?
The comments on lines 115-117 about adolescents and children is unnecessary and should be removed. If the WHO definition of child is going to be used (including age 19), then just use that definition, no need to introduce more confusion by switching over to the Convention on Rights of the Child.
In general, I am not sure what the long list of definitions on page 3 adds to the manuscript. These are relatively well-accepted terms already. If the editors are interested in shortening the manuscript, much of this could be removed.
The inclusion criteria say migrants to all areas of the world -- nothing about only to low-resource settings. Then in Figure 1, I see that that 5 articles were excluded because they were carried out in high-income countries. But then in Table 1, I see that several studies that were carried out in high-income countries are again included (USA, Germany, Canada, etc.). This fundamental error in rigorously applying the Methods unfortunately undercuts the Results and Discussion and makes the paper unacceptable in its current form. The search needs to be done again and the paper rewritten with the papers identified that meet the criteria the authors proposed in their study design.
Author Response
Here is our point-by-point reply to the reviewer's comments.
- This is a systematic review evaluating the food insecurity and nutritional status of vulnerable children during migration. This is an interesting question and one that would have been good to have in the nutrition community. Unfortunately a significant methodological flaw makes the paper unacceptable in my mind for publication in its current form. The authors could consider repeating the literature search with more proper attention to inclusion and exclusion criteria, etc. A systematic review that is not done systematically and rigorously is really of little use to the field.
The inclusion of migrants but the exclusion of refugees is subtle when considering issues of food insecurity and malnutrition. I can understand the motivation and rationale for keeping the two populations separate, as they have different risk factors and outcomes, but this difference may be lost on casual readers. I would recommend that in the Abstract and Introduction, this should be made much more explicitly clear and emphasized, including an explanation of the rationale and examples of how these two populations may reasonably assumed to have differing risk factors and outcomes.
Reply: thank you for underlining this important point. Indeed, refugees are a different population with some peculiarities and that is why we decided to limit our research to migrants. We have added an explanation of the main differences and of our choice at the end of the introduction (lines 101-108) and also, more briefly, in the abstract.
- Given that refugees are to be excluded in this review, then why was “refugee*” included in the search string?
Reply: Thank you for this comment. When we built up the search string (using the help of a methodologist) we decided to take into account all the words (synonyms, abbreviations, etc) which could have been potentially been involved in describing our PICO question. Refugee was, obviously, one of the principal words taken into account, and one of the most important in retrieving papers. AS you know, sometimes papers (eg Lane et al, 2018) analyse together migrants and refugees. In these cases, the exclusion of the world “refugee” may have excluded papers that simultaneously analyzed both migrants and refugees, and therefore would have narrowed (specifying) our research and causing a loss in part of the information/relevant data.
- The comments on lines 115-117 about adolescents and children is unnecessary and should be removed. If the WHO definition of child is going to be used (including age 19), then just use that definition, no need to introduce more confusion by switching over to the Convention on Rights of the Child.
Reply: you are right, the definition of adolescent is redundant to this paper and we have removed it.
- In general, I am not sure what the long list of definitions on page 3 adds to the manuscript. These are relatively well-accepted terms already. If the editors are interested in shortening the manuscript, much of this could be removed.
Reply: we think it is important to keep the definitions of these terms, because they can help the “casual” reader to understand the topic. We think that an option might be to move them to the supplementary materials. We ask the editor to tell us which option is better for the journal and we can change the paragraph consequently.
- The inclusion criteria say migrants to all areas of the world -- nothing about only to low-resource settings.
Reply: we thank the reviewer for highlighting this point. We have corrected the text into “migrants from low- to high-resource settings” (lines 131-132) and consequently introduction l. 86)
- Then in Figure 1, I see that that 5 articles were excluded because they were carried out in high-income countries. But then in Table 1, I see that several studies that were carried out in high-income countries are again included (USA, Germany, Canada, etc.). This fundamental error in rigorously applying the Methods unfortunately undercuts the Results and Discussion and makes the paper unacceptable in its current form. The search needs to be done again and the paper rewritten with the papers identified that meet the criteria the authors proposed in their study design.
Reply: We are particularly grateful for this comment, which allows us to improve the quality of our paper. We are sorry that the short specification of HIC (High Income Countries) could have mislead the reader from the real purpose of that specification in the flowchart. In fact, those 5 papers you rightly highlighted, were either carried out only in population migrating between different regions in highly developed countries, or between countries which were both already highly developed. We acknowledge that the caption in the flowchart could have been misleading, so now we have corrected it in order to be clearer for the readers.
Reviewer 3 Report
This paper addressed an important question in nutrition. The process of identifying relevant literature and evaluation work is extensive and careful
Major comments:
All the finding should include CI and P-values so that the readers can benefit from reading meta-analysis. I am not expert in nutrition, but I think the work need to make some statistical analysis to get meta-analysis results or at least give the reason why no statistical analysis.
Author Response
Here is our point-by-point reply to the reviewer's comments.
This paper addressed an important question in nutrition. The process of identifying relevant literature and evaluation work is extensive and careful
Major comments:
- All the finding should include CI and P-values so that the readers can benefit from reading meta-analysis.
Reply: Thank you for your comment. We have added the available statistical information to the text all along the “Results” section.
- I am not expert in nutrition, but I think the work need to make some statistical analysis to get meta-analysis results or at least give the reason why no statistical analysis.
Reply: As we performed a sensitivity analysis for the outcomes of interest and the analysis indicated that results were not solid enough to perform a meta-analysis, we decided to conduct only a qualitative evaluation of evidences (systematic review). Systematic review methods have been subject to considerable discussion and debate especially regarding the selection of studies to include or exclude from review. The traditional approach believes that they must deal with all studies regardless of the quality. The critical evaluation approach aims to include only studies that meet a predetermined threshold of quality. Adherence to either approach may impose serious limitations. As an alternative, some systematic reviewers advocate an intermediate approach that considers the merits of both positions (Slavin, R. E. (1987). Best-evidence synthesis: Why less is more. Educational Researcher, 16, 15–16.). We totally agree that the results of a meta-analysis are influenced by the quality of the primary studies included. Performing our systematic review (without a meta-analysis), we chose an intermediate approach to include as many studies as possible without jeopardizing internal validity.
To note, a quality analysis of studies included in the systematic review was performed, and results were reported in the revised version of the manuscript (Table S1). We strongly believe that our approach is correct when the number of studies fitting the inclusion/exclusion criteria are not sufficient to perform a meta-analysis. Inclusion of all eligible studies allows an unbiased overview of the state of knowledge, while qualitative analysis of included studies justifies the final critical evaluation of evidences in the systematic review.
Round 2
Reviewer 2 Report
Will need significant English editing.
Author Response
"Will need significant English editing."
One of the authors (Sugitha Sureshkumar) is an English-native speaker and the paper was written with her language supervision. All the versions had been checked for English language before submission. She has rephrased some sentences that could have been said better.
If the reviewer thinks that more corrections are to be made, we kindly ask him/her to highlight the paragraphs/points in which he/she considers to edit the language.